# Direct synthesis of bicyclo[1.1.1]pentane (BCP) boronates from carboxylic acids

Yongchen Wang[1], Jess C. Tang[1], Gang Wu [2] & Julian G. West [1] ✉

Bicyclo[1.1.1]pentane (BCP) boronic esters are crucial intermediates for accessing BCP-containing drugs with improved pharmacokinetic profiles, yet their synthesis typically relies on pre-formed redox-active esters derived from carboxylic acids. Here we report a general, single-step method for the direct conversion of carboxylic acids into BCP boronic esters. Upon irradiation of carboxylic acids with [1.1.1]propellane and bis(pinacolato)diboron ($B_2pin_2$) in dimethyl sulfoxide (DMSO), BCP boronates are obtained in good yields, which are further enhanced by the addition of an iron catalyst. Mechanistic studies suggest that photolytic cleavage of a $B_2pin_2$–DMSO complex initiates decarboxylation via hydrogen atom transfer (HAT), while iron catalysis enables a parallel ligand-to-metal charge transfer (LMCT) pathway. This synergistic HAT/LMCT process displays broad substrate scope and remarkable functional group tolerance. Additionally, BCP analogs of two approved drugs, butenafine and buclizine, have been readily synthesized, underscoring the potential of this dual HAT/LMCT paradigm to reshape strategies in synthetic chemistry and drug discovery.

As a bioisostere of the para-substituted benzene ring, 1,3-disubstituted BCPs have been increasingly incorporated in the fields of medicinal chemistry, organic chemistry, material chemistry, and pharmaceuticals, significantly improving the pharmacokinetic and physico-chemical properties of drugs[1–10]. Toward synthesizing these BCP-containing drug candidates and other bioactive molecules, 1,3-disubstituted BCP boronates have notable potential as synthetic building blocks due to the versatility of the boronate functional group, allowing myriad downstream functionalizations. Despite this promise, early methods to synthesize 1,3-disubstituted BCP boronates developed by Aggarwal, Uchiyama, and Walsh have lengthy synthesis steps, require the participation of organometallic reagents, and/or have limited substrate applicability, presenting a barrier to accessing these key motifs[11–13]. To address these issues, the Molander group reported visible light-induced multicomponent reactions (MCRs) of [1.1.1]pro-pellane for the one-step synthesis of BCP boronates using either redox active esters (RAEs) derived from carboxylic acids or organohalides as the alkyl fragment precursors (Fig. 1A)[14]. The use of RAEs was a notable advance, as carboxylic acids are among the most available and

sustainable alkyl fragment donors available to chemists[15,16]. However, a major limitation of this method is the requirement to pre-activate these carboxylic acids as N-hydroxyphthalimide (NHPI) RAEs, significantly reducing the atom and step economy of the synthesis and complicating the product stream. Therefore, direct decarboxylative borylation of carboxylic acids with [1.1.1]propellane to prepare a series of BCP boronates would represent an ideal approach to 1,3-disubstituted BCP boronates, allowing access to these critical fragments in one step from commodity precursors with high atom economy. Herein we report the realization of this aspirational goal: a general approach to 1,3-difunctionalized BCP-Bpins via the direct decarbox-ylation of various carboxylic acids, utilizing an iron catalyst, $Cs_2CO_3$, and DMSO solvent under light irradiation, allowing direct and rapid access to a broad scope of 1,3-disubstituted BCP boronates (Fig. 1D). Intriguingly, the desired products can still be obtained with a satis-factory yield in the absence of an iron catalyst and base, revealing a powerful photoreactivity for $B_2Pin_2$ in DMSO and supporting the reaction occurring via an unusual O−H HAT-mediated decarboxylation mechanism (Fig. 1D). This mechanism is complementary to common

[1]Department of Chemistry, Biosciences Research Collaborative, Rice University, Houston, TX, USA. [2]Department of Internal Medicine, University of Texas-McGovern Medical School, Houston, TX, USA. ✉e-mail: jgwest@rice.edu

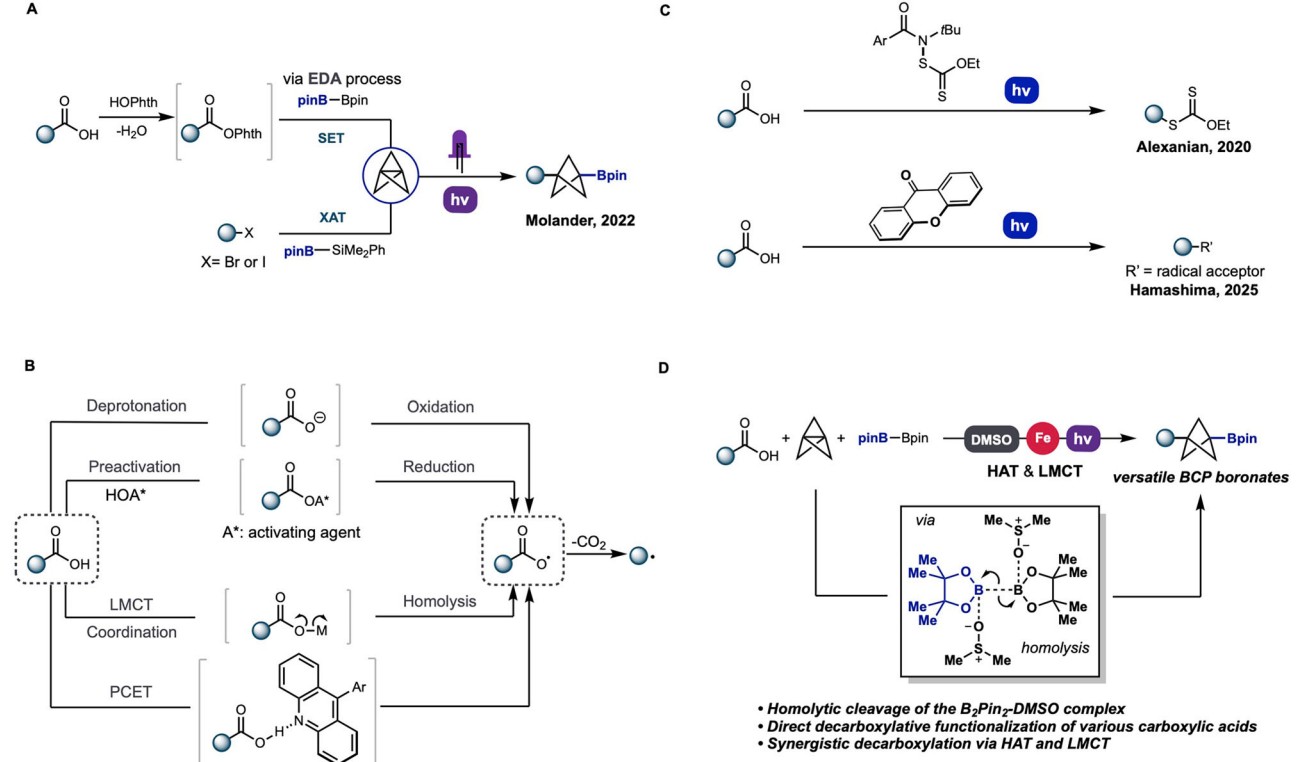

**Fig. 1 | Background and project design. A** Previous one-step method to access 1,3-disubstituted BCP boronates. **B** General approaches to decarboxylation of carboxylic acids. **C** Reported direct decarboxylation of carboxylic acid via O-H HAT.

**D** This work: Direct decarboxylation of carboxylic acids by O–H HAT in synergy with LMCT to enable synthesis of a wide range of BCP boronates.

deprotonation/oxidation or RAE preactivation[17–23], ligand-to-metal charge transfer (LMCT)[24–28], and proton-coupled electron transfer (PCET)[29–32] decarboxylation approaches (Fig. 1B) and has only been observed using specialized *N*-xanthylamides[33] or photoactivated aryl ketones to perform O–H HAT (Fig. 1C)[34]. Our studies support in situ photolytic generation of •OBpin from $B_2Pin_2$ in DMSO permits this same HAT-enabled radical decarboxylation mechanism, with the inclusion of iron photocatalyst enabling more efficient reaction via a parallel synergistic LMCT decarboxylation.

## Results

Our investigation of the decarboxylative borylation reaction commenced with 1-methyl-1-cyclohexane carboxylic acid (2j) as the model substrate for reaction optimization. Irradiating the mixture of acid (2j), [1.1.1]propellane (1) and $B_2pin_2$ in the presence of $Fe(NO_3)_3 \cdot 9H_2O$ as catalyst, $Cs_2CO_3$ as base, and DMSO as solvent at room temperature for 20 h allowed us to obtain the desired product (3j) in a high 98% yield (Table 1, entry 1), leading us to select these as standard conditions. In order to better study the reaction character, we next sought to change some key reaction parameters to see how the reaction was affected. BCP boronate product was not detected in the absence of light (Table 1, entry 2); however, we still obtained the desired product in satisfactory yield (70%) in the absence of iron catalyst and with 1.0 equivalent of base (Table 1, entry 3), demonstrating iron to be beneficial, albeit not required, for product formation. Excluding base from the standard conditions led to borylated product in a reduced 58% yield (Table 1, entry 4), showing this reagent to again be beneficial but not required for product formation. Intriguingly, exclusion of both iron catalyst and base allowed product formation in high yield (83%, 86%) (Table 1, entry 5), enabling access to 1,3-disubstituted BCP boronates in a simple reaction system and suggesting an unanticipated

decarboxylation mechanism. Finally, replacing DMSO with MeCN or DMA dramatically reduced product formation to 31% yield and 14% yield, respectively (Table 1, entry 6, 7), indicating that DMSO plays a pivotal role in the success of the reaction. It is worth mentioning that when the standard reaction was scaled up to 1 mmol, an isolated yield of 85% was still achieved. (see Supplementary Information section 3.4).

With optimized conditions in hand, we began to explore the effect of different carboxylic acid substrates on the reaction. To our delight, we found this substituted BCP boronates synthesis could be applied to a wide range of carboxylic acids bearing diverse functionality, including primary, secondary, tertiary, and benzylic acids (Fig. 2). Beginning with primary alkyl carboxylic acids, saturated acids (2a, 2b) gave the desired products (3a, 3b) in 62% and 81% yield separately. An alkyne substituent in carboxylic acid (2c) was also shown to be compatible, providing the BCP boronate product (3c) in 70% yield. Engaging an acid with an adjacent cyclopropyl functional group (2 d) provided the ring-opening product (3 d) in high yield (89%), supporting a radical mechanism. Finally, acid bearing phenoxy ether (2e) provided corresponding product (3e) in moderate yield (46%), supporting compatibility of ethers and arenes. Happily, secondary carboxylic acids featuring electron-withdrawing (3 f, 3 g, 71%, 71%) and electron-donating (3 h, 3i, 82%, 81%) substituents were also accommodated, giving us the corresponding product in high yield. Tertiary carboxylic acids exhibited notable generality, with those featuring *tert*-butyl (2k) and methyl or fluorine-substituted *tert*-butyl moiety (2 l, 2 m) producing the desired product in good yield (3k-3m, 62–88%). In addition, tertiary carboxylic acids containing *tri*- (cyclopropane, 2n), *hexa*-cyclic ring (tetrahydropyran, 2o) and *tetra*- (oxetane, 2p) substituents were also well compatible and provided moderate to high yield of desired products (3n-3p, 32–80%). Tertiary carboxylic acid with methyl piperidine core (2q) worked very well, delivering the corresponding

**Table 1 | Optimization of direct decarboxylation conditions**

| entry | Deviation from the standard conditions | NMR yield[a] |
|---|---|---|
| 1 | none | 98% |
| 2 | No light | N.D. |
| 3 | No iron, Cs₂CO₃ (1.0 equiv.) | 70% |
| 4 | No Cs₂CO₃ | 58% |
| 5 | No iron and no Cs₂CO₃ | 83%, 86%[b] |
| 6 | MeCN instead of DMSO | 31% |
| 7 | DMA instead of DMSO | 14% |

Reaction conditions were as follows: [1.1.1]propellane 1 (0.1 mmol, 1.0 equiv.), carboxylic acid 2j (0.2 mmol, 2.0 equiv.), B₂pin₂ (0.3 mmol, 3.0 equiv.), Fe(NO₃)₃·9H₂O (0.01 mmol, 0.1 equiv.), Cs₂CO₃ (0.02 mmol, 0.2 equiv.) and solvent (0.1 M), 20 h, rt, N₂, 390 nm Kessil purple LED. [a]The $^1$H NMR yield is determined by using 1,2-dichloroethane as an internal standard; [b]The reaction in the absence of iron and Cs₂CO₃ was performed twice and delivered the product in 83% and 86% NMR yield, respectively.

product (3q) in 83% and demonstrating compatibility with N-containing heterocyclic scaffolds. Applying fluorinated BCP acid (2r) to the reaction system results in the formation of BCP boronate with the bis-BCP staffane scaffold (3r) in 65% yield, allowing access to this intriguing dyad and demonstrating the amenability of bicyclic ring carboxylic acids to the method. Indeed, tertiary carboxylic acids incorporating a strained ring, such as noradamantane (2 s) and dimethyl substituted adamantane (2t) were also reactive, giving us the desired product (3 s, 3t) in 56% and 87% yield, respectively. Further, tertiary carboxylic acids bearing aryl ether phenoxy (2 u) or chlorine-substituted phenoxy group (2 v) were able to deliver the desired product with high yield (3 u, 3 v, 83%, 73%), allowing incorporation of this common pharmacophore.

Having demonstrated the amenability of alkyl and alpha-heteroatom carboxylic acids, we next sought to determine whether activated arylacetic acids would also participate in the reaction. To our delight, this substrate class functioned effectively, with primary (3w, 60%) and secondary (3x-3aa, 49–61%) aryl acetic acids providing moderate to good yield of desired products. Approved drug molecules bearing arylacetic acids, such as fenoprofen (3ab, 63%) and ibuprofen (3ac, 57%) similarly functioned well, showing this moiety to be amenable for late stage functionalization. Cyclic arylacetic acids possessing cyclobutane and cyclohexane moieties were suitable for the reaction as well, offering us the desired products (3ad, 3ae) with 51% and 50% yield, respectively and again permitting cyclic dyads to be synthesized directly. On top of these examples, tertiary arylacetic acids were also amenable to the conditions (3af, 3ag, 64%, 54%), including cyclic arylacetic acids containing cyclopropane and cyclobutene substituents (3ah-3aj, 54–80%), although the reaction of the acid bearing cyclopentane (2ak) was slightly less efficient (3ak, 30%). Notably, gemfibrozil, a drug to treat high cholesterol and triglyceride levels, reacted smoothly, delivering the desired BCP boronate (3al) in high yield (82%).

Finally, the potential of the chemistry to operate in complex natural products was demonstrated by reaction of abietic acid and dehydroabietic acid, which both afforded the corresponding boronates in good yield (3am, 3an, 63%, 74%). Last but not the least, this methodology also applied to carboxylic acid bearing free hydroxyl group and amino acid (Boc-Pro-OH) and rendered the desired product (3ao, 3aq, 69%, 65%) in good yield. The oxidation of hydroxyl boronate (3ao) afforded the BCP aldehyde (3ap) in 47% yield over two steps.

To demonstrate the utility of our simple and efficient method to access diverse BCP boronates, we next investigated how these products can enable incorporation of BCP bioisosteres into drug candidates. We identified two approved drugs, butenafine and buclizine, as targets whose BCP surrogates could be unlocked using our BCP boronate fragments (Fig. 3)[35,36]. The synthesis of the butenafine bioisostere analog began with homologation of *tert*-butyl substituted BCP boronate (3k), followed by sodium perborate mediated oxidation to give primary alcohol (3 kb). The subsequent Dess-Martin oxidation afforded the corresponding aldehyde (3kc), which was then submitted to reductive amination with amine (3kd) as an amine source, allowing synthesis of the desired BCP-containing butenafine (3ke) in 18% overall yield over four steps. Similarly, reductive amination reaction of the same aldehyde (3kc) with the other amine source (3kf) allowed the buclizine bioisostere analog (3 kg) to be accessed in 17% overall yield over four steps. These preliminary examples demonstrate how this method can be rapidly deployed in medicinal chemistry campaigns to open the door to the synthesis of myriad drug candidate molecules bearing a BCP substituent.

To acquire preliminary mechanistic insights into our synergistic decarboxylation protocol, we initially aimed to investigate whether the reaction is mediated by a radical process (Fig. 4). Addition of 1.0 equivalent of radical scavenger 2,2,6,6-tetramethyl-1-piperidinyloxy (TEMPO) in the standard system resulted in the formation of TEMPO

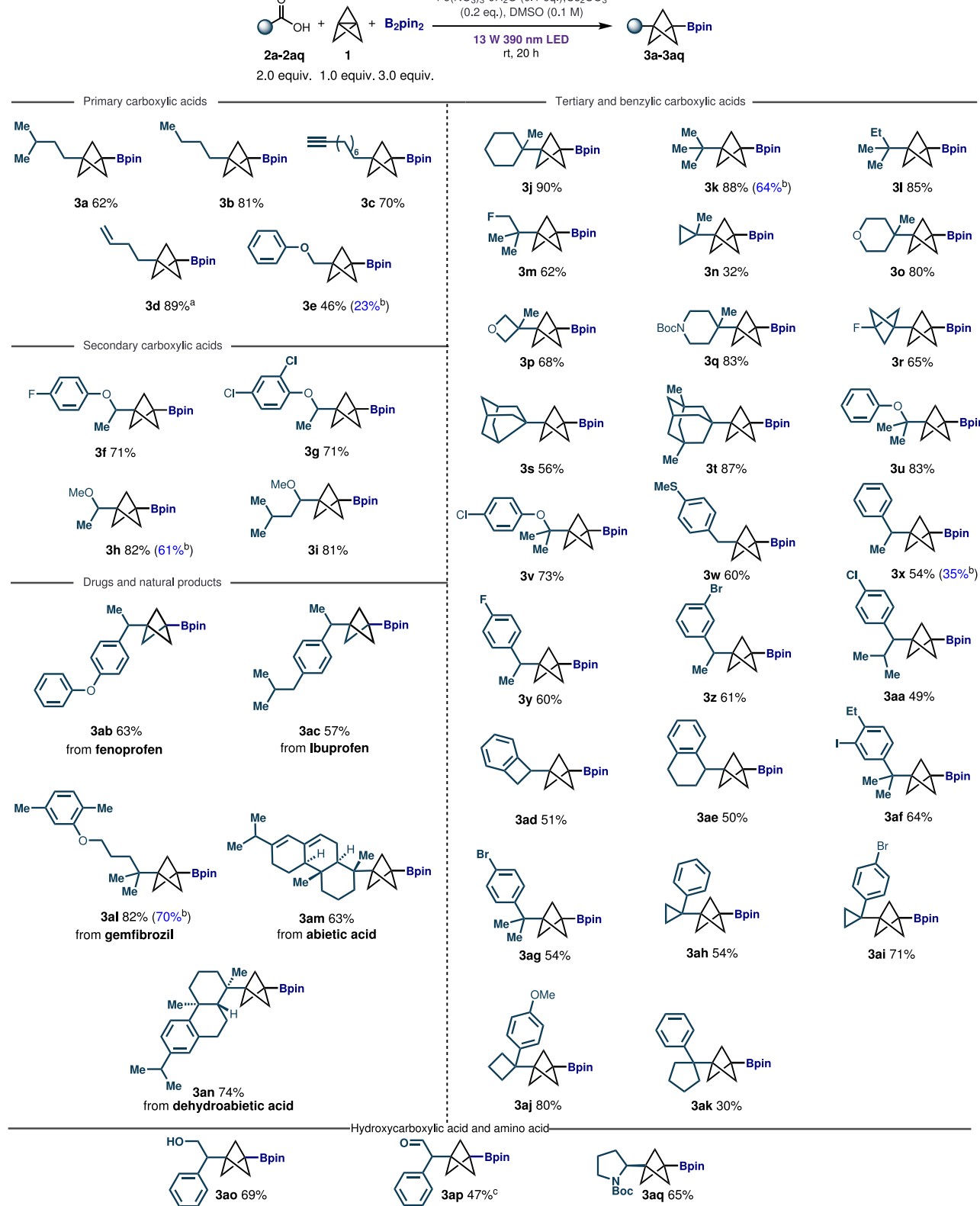

**Fig. 2 | Scope of BCP boronates synthesis via direct decarboxylation.** Reaction conditions were as follows: [1.1.1]propellane (0.1 mmol, 1.0 equiv.), carboxylic acids (0.2 mmol, 2.0 equiv.), B₂pin₂ (0.3 mmol, 3.0 equiv.), Fe(NO₃)₃·9H₂O (0.01 mmol, 0.1 equiv.), Cs₂CO₃ (0.02 mmol, 0.2 equiv.) and DMSO (0.1 M), 20 h, rt, N₂, 13 W

390 nm Kessil purple LED. Yields are isolated yields; [a]3d was obtained from 2-cyclopropylacetic acid (Radical clock experiment); [b]The yields in the parentheses are the isolated yields of the reactions under iron-free and base-free conditions; [c]The synthesis of aldehyde (3ap) is from the oxidation of alcohol (3ao).

**Fig. 3 | Synthetic pathway to bioisosteres of butenafine and buclizine.** Direct decarboxylation allows access to two bioisosteres from a common BCP boronate reagent.

adduct product, which was detected by HRMS (Fig. 4A), a result consistent with radical intermediates. To further support the radical pathway of the reaction mechanism, a radical clock experiment was performed with cyclopropyl carboxylic acid (2d), and the expected ring-opening product (3d) was obtained in high yield (Fig. 4B), again supporting decarboxylation to form a carbon-centered radical intermediate. Additionally, no trace of product was observed in the absence of light, suggesting a photochemical process. With this evidence for radical formation, we next sought to explore how the radical was generated under not only our standard reaction conditions, but also conditions in the absence of an iron catalyst and base. A recent report from Hu and coworkers showed that a mixture of $B_2cat_2$ and DMSO allows in situ generation of a 1:1 $B_2cat_2$-DMSO complex with unique reactivity[37], prompting us to consider whether an analogous 1:1 or 1:2 $B_2pin_2$-DMSO complex might form under our conditions. If such a complex were to be photoactive, the B-B bond might undergo photolytic cleavage to afford the •OBpin radical which could serve as an O-H HAT reagent for carboxylic acid radical decarboxylation and dimethylsulfide (DMS) as a stoichiometric byproduct. To test this possibility, we measured UV-vis spectra for solutions of $B_2Pin_2$ in DMSO as well as mixtures of carboxylic acid, $B_2Pin_2$, and [1.1.1]propellane in a 1:1:1 mixture ("HAT cycle") and a 2:3:1 mixture ("HAT condition," the same conditions as those shown in Table 1, Entry 5) and compared their absorbance (Fig. 4C). As expected, we observed significant absorbance for our reaction mixture both in 1:1:1 and 2:3:1 reactant ratios, with the strongest absorbance for the standard "HAT condition". Interestingly, we did not observe significant absorbance of $B_2pin_2$ in DMSO in the absence of the other reaction components, suggesting a synergistic interplay of multiple reagents for efficient photochemistry. With that said, we found that irradiation of a solution of $B_2pin_2$ in DMSO lead to the formation of DMS which was detected by HRMS and its characteristic odor (Fig. 4D, see Supplementary Information section 3.5 for additional information). Bpin-OH was not detected by HRMS in the DMSO solution of $B_2Pin_2$, potentially due to a lack of suitable hydrogen atom sources. No DMS was detected in the absence of light. DMS was also detected after irradiation of $B_2Pin_2$ in the presence of carboxylic acid (2j) alone; acid (2j) and [1.1.1]

propellane; and acid (2j), [1.1.1]propellane, iron catalyst, and base, suggesting this photolytic process to be relevant under the reaction conditions. All reactions containing carboxylic acid also formed Bpin–OH, suggestive of the proposed O−H HAT abstraction process. No DMS was formed in the absence of light for any of these conditions. Interestingly, replacing DMSO with different equivalents of diphenyl sulfoxide under iron and base free conditions still yielded the desired product (3j) from 32% to 46% yield, and diphenyl sulfide was also obtained in reasonable amounts (Fig. 4E, see Supplementary Information section 3.5 for additional information). No desired product (3j) was observed in the absence of sulfoxide source and this indicated sulfoxide is a key factor in the success of the reaction. EPR experiments were also carried out with addition of spin-trap reagent 5,5-dimethyl-1-pyrroline N-oxide (DMPO) (Fig. 4F). After irradiating a solution of $B_2pin_2$ in DMSO for 1 h, the solution was subjected to EPR analysis and, to our delight, an oxygen-centered radical was observed as indicated by the hyperfine splitting constants, $a_N = 13.7$ G and $a_{\beta-H} = 11.7$ G, that are typical of DMPO adduct with an oxygen-centered radical (reaction a)[38]. The oxygen-centered radical most likely originates from a $B_2pin_2$-DMSO complex, suggesting the formation of •OBpin (see Supplementary Information section 3.5 for EPR spectra and detailed results analysis). EPR analysis of a solution of carboxylic acid and $B_2Pin_2$ (reaction b) was more complicated, suggesting both oxygen centered and carbon centered radicals were captured by DMPO (see Supplementary Information section 3.5 for EPR spectra and detailed results analysis). The radicals captured in reactions containing carboxylic acid, [1.1.1]propellane, and $B_2Pin_2$ in DMSO (reaction c) and the same mixture with catalytic iron (reaction d) were mainly carbon-centered radicals, as indicated by the hyperfine splitting constants $a_N = 14.6$ G and $a_{\beta-H} = 21.3$ G and $a_N/a_{\beta-H} = 0.69$ that are typical of DMPO adducts with carbon-centered radicals[39,40] (see Supplementary Information section 3.5 for EPR spectra and detailed results analysis). These EPR data support the ability of $B_2Pin_2$/sulfoxide mixtures to photolytically generate reactive oxygen species and achieve radical decarboxylation in the presence of carboxylic acids.

Comparison of carboxylic acid containing reactions supports that the decarboxylation is enhanced by addition of iron

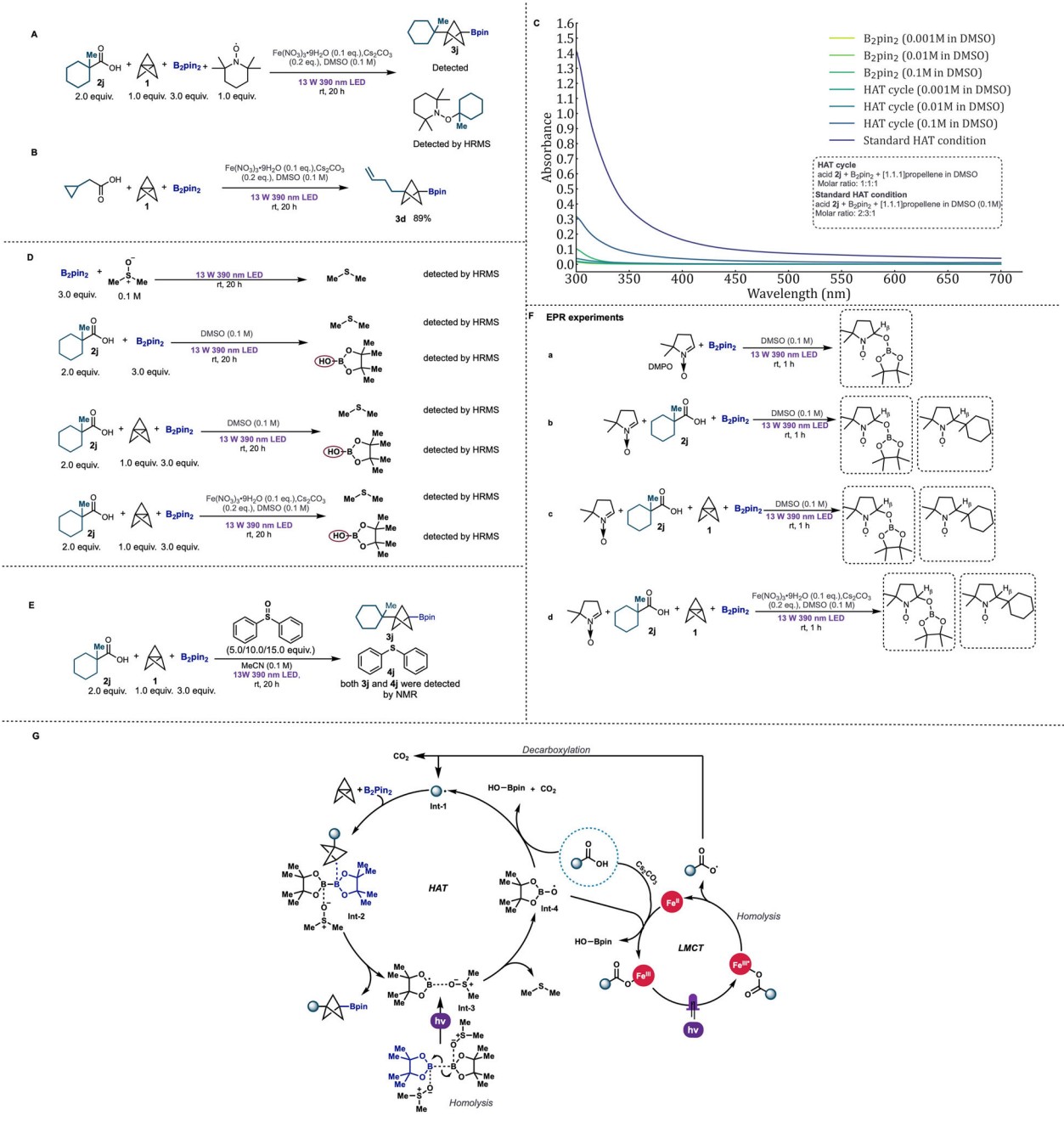

**Fig. 4 | Mechanistic studies. A** TEMPO involved radical scavenger experiment. **B** Radical clock experiment. **C** UV-vis spectra. **D** HRMS detection results. **E** DMSO replacement experiments. **F** EPR experiments. **G** Proposed reaction mechanism.

salt and base, as a significantly larger carbon-centered radical EPR signal for reaction d compared to the same signals for reactions b and c (see Supplementary Information section 3.5 for EPR spectra and detailed results analysis). This enhanced decarboxylation is further supported by the higher yield of the desired product (3j, 98%) in the presence of iron catalyst. Taken together, these EPR and additional experimental results are consistent with an LMCT-mediated decarboxylation process contributing productively to the reaction.

Based on the above experiments, we proposed the following decarboxylation mechanism mediated by $B_2pin_2$-DMSO HAT and in synergy with iron-mediated LMCT (Fig. 4G). A $B_2pin_2$-DMSO complex could undergo homolysis under visible light irradiation to give Int-3, which could release OBpin radical Int-4 with the extrusion of DMS.

Next, the hydrogen atom of a carboxylic acid could be abstracted by the OBpin radical, resulting in the formation of R• radical with the release of $CO_2$. [1.1.1]Propellene could then be attacked by the R• radical, followed by $B_2pin_2$, resulting in formation of the desired substituted BCP boronates with co-production of Int-3, which would continue to participate in the HAT-mediated catalytic cycle. In a parallel process, an LMCT-mediated catalytic cycle could begin with carboxylic acid deprotonation and coordination to the iron catalyst. Irradiating the iron carboxylate complex under 390 nm irradiation would then promote the LMCT process, producing the R• radical which could then attack [1.1.1]Propellene, intercepting product formation as described above. The generated OBpin radical, could then reasonably serve as an oxidant to return $Fe^{II}$ to $Fe^{III}$ and enable catalysis.

## Discussion

We have demonstrated a general synthetic approach to a range of diverse BCP boronates with a simple and one-step setup, which opens up a direct and efficient route to these critical synthetic fragments in high yield. Our method functions via direct decarboxylative borylation of feedstock carboxylic acids in a single step without the need for pre-functionalization of carboxylic acids to corresponding NHPI RAEs, presenting high step and atom economy. Furthermore, we successfully converted BCP boronates synthesized using our method into the bioisostere of approved drugs, demonstrating how this method can unlock more potential drug molecules for the pharmaceutical industry. Additionally, the reaction proceeds via a unique direct photo-decarboxylation process enabled by combination of $B_2pin_2$ and DMSO, likely via direct O–H HAT, and can be further improved through addition of a synergistic iron LMCT catalyst. This study intoduces this reactivity using the $B_2pin_2$/DMSO reagent combination and suggests that this distinctive activation mode will enable a wide range of high-value transformations in the future.

## Methods

### General procedure for direct synthesis of BCP boronates from carboxylic acids

To a solution of carboxylic acids (0.2 mmol, 2.0 equiv.), bis(pinacolato) diboron (0.3 mmol, 3.0 equiv.), $Cs_2CO_3$ (0.02 mmol, 0.2 equiv.) and $Fe(NO_3)_3 \cdot 9H_2O$ (0.01 mmol, 0.1 equiv.) in DMSO (1.0 mL) under $N_2$ atmosphere was added [1.1.1]propellane (0.1 mmol, 1.0 equiv.) The punctured holes of the vial cap were sealed with vacuum grease and electric tape/parafilm for better air-tight protection. The reaction mixture was sonicated for about 60 s until it was almost clear. The reaction mixture was then placed under 390 nm Kessil® light (25% intensity, 13 W) with a cooling fan. After stirring at room temperature for 20 h, the reaction mixture was quenched with water. The layers were separated and the aqueous layer was extracted with $Et_2O$. The combined organic layers were washed with brine, dried over $Na_2SO_4$, filtered and concentrated under reduced pressure (The resulting crude residue was passed through a short silica gel plug prepared in a glass pipette and eluted with $CH_2Cl_2$ to remove iron salts, affording material suitable for crude $^1H$ NMR analysis when required and 1,2-dichloroethane was used as an internal standard for $^1H$ NMR yield). The crude residue was purified by flash column chromatography to give corresponding BCP boronic esters.

## Data availability

The authors declare that all the data supporting the findings of this research are available within the article and its Supplementary Information. Additionally, all data are available from the corresponding author upon request.

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

## Acknowledgements

We acknowledge financial support from CPRIT (RR190025), NIH (R35GM142738), the Welch Foundation (C-2085), RCSA (CS-CSA-2023-007), and Eli Lilly (A-36829). J.G.W. is a CPRIT Scholar in Cancer Research. Dr. Ian M Riddington and Dr. Jongdoo Lim (UT Austin Mass Spectrometry Facility) are acknowledged for assistance with mass spectrometry analysis.

## Author contributions

Y.W. and J.G.W. designed the project. Y.W. and J.C.T. performed the non-EPR experiments. Y.W., J.C.T., and J.G.W. wrote the manuscript. Y.W. and G.W. performed the EPR experiments. J.G.W. directed the project. All authors interpreted the results in the manuscript.

## Competing interests

The authors declare no competing interests.
