## [Transparent Peer Review file · Nature Communications]

Direct synthesis of bicyclo[1.1.1]pentane (BCP) boronates from carboxylic acids

Corresponding Author: Professor Julian West

Version 0:

Reviewer comments:

Reviewer #1

(Remarks to the Author)

In this manuscript, the authors present a straightforward and practical photoredox method for synthesizing BCP boronates from readily available aliphatic carboxylic acids. Under irradiation, a mixture of carboxylic acids, [1.1.1]propellane, and B2Pin2 in DMSO efficiently affords the target BCP boronates in good yields. Mechanistically, the authors propose a novel pathway involving photolytic cleavage of a B2Pin2–DMSO complex, which drives radical decarboxylation via hydrogen atom transfer. The addition of an iron catalyst further enhances the reaction yields by promoting decarboxylation through a ligand-to-metal charge transfer process. A wide range of carboxylic acids, including primary, secondary, and tertiary alkyl carboxylic acids are compatible with this system, without the need for prefunctionalization of the carboxylic acids. Thus, I recommend the publication of this work in Nature Communications after addressing the following comments.

- 1) Please comment on the scalability of this reaction system.
- 2) The manuscript would be strengthened by a more detailed discussion on the limitations of the substrate scope and the functional group tolerance within the alkyl carboxylic acids. For the practical utility of this method, it is crucial to understand its compatibility with common functional groups. Key questions include: are free alcohols, amino groups, and heteroaromatics like pyridines tolerated? Moreover, the applicability of α -amino acids, a valuable class of substrates, would be of particular interest to the community.
- 3) Please carefully check the HMRS data in the supporting information, with particular attention to the calculated masses. For instance, regarding products 3a and 3b, the anticipated calculated mass values for 3a as $[M+H]^+$ ($C_{16}H_{30}BO_2^+$) and for 3b as $[M+H]^+$ ($C_{15}H_{28}BO_2^+$) should be 265.2333 and 251.2177, respectively.

Reviewer #2

(Remarks to the Author)

West et al. developed a new method for synthesizing BCP boronate esters. This reaction allows carboxylic acids to be used directly without converting them to redox-active esters, making it highly valuable in synthetic chemistry. Under iron catalyst conditions, the addition reaction to [1.1.1]-propellane proceeds smoothly, regardless of the carboxylic acid series, and the desired boronate esters can be obtained in good yields. Furthermore, tolerance for heterofunctional groups and aromatic rings has been confirmed.

The reaction mechanism contains novel and interesting insight. It is intriguing that the reaction proceeds without the addition of an iron catalyst. It has been previously discussed that the (pin)BO oxy-radical is generated when a sulfoxide-B2pin2 complex is photoexcited (ref 37). Also, the detection of DMS and (pin)BOH in Fig. 3 experimentally supports the generation of the (pin)BO oxy-radical. However, it is puzzling that no molecules derived from the (pin)BO oxy-radical were detected in the first reaction of Fig. 3d.

The mechanistic proposal is made based on the detection of DMS and pinBOH, but no quantitative discussion is provided. Considering that the diboron has a reducing nature and DMSO is oxidative, it is insufficient to propose Fig. 3e for the HAT cycle based on their detection alone. If $Ph_2S=O$ could be used instead of DMSO, quantitative discussion would be possible. Furthermore, the fact that the carbon radical derived from the carboxylic acid preferentially adds to BCP and not to diboron has not been well elucidated. It is desirable to provide an explanation, citing references as necessary.

A novel aspect is that the (pin)BO oxy-radical causes HAT selectively for carboxylic acid O-H bond, but unfortunately there is no discussion for the reason. The starting carboxylic acid and the product contain C-H bonds with significantly smaller BDEs than the carboxylic acid O-H bond, and yet high selectivity for carboxylic acid and (estimated) BDE value of (pin)BO-H bond are of great interest. The changes in the UV-Vis absorption spectrum (Fig. 3c) strongly suggest an interaction between the carboxylic acid and the diboron, which might provide a clue to explaining the chemoselectivity. A clear explanation for this, based on experimental or theoretical data, is necessary for publication in Nature Commun which requires a high level of academic merit.

In this context, it is debatable that their unique HAT cycle applies only to carboxylic acid 2j and the generality of the catalytic system has not been investigated at all.

This study reports a synthetically useful reaction and thus I am positive about publication in this journal. However, there are many ambiguities regarding the novel HAT cycle, which I think requires extensive revisions. Other minor errors that I have noticed are as follows:

Abstract: line 15: direct synthesis of BCP. "of" is missing.

Page 2: line 49: N-hydroxyphthalimide. "H" should be lowercase.

P4, line 70: 1,3-disubstituted. "di" is missing.

P13, line 226: "the proton of a carboxylic acid" should be corrected to "the hydrogen atom of a carboxylic acid O-H."

P14, Fig 3c: Legends are too small. Please use the picture with much higher resolution.

Reviewer #3

(Remarks to the Author)

West and co-workers report a new synthetic approach to bicyclo[1.1.1]pentane (BCP). BCP has recently attracted significant attention as a bioisostere of para-substituted benzene, and two main strategies have been widely explored: those using [1.1.1]propellane and those using bicyclo[1.1.0]butane as precursors. The present manuscript falls into the former category. The authors demonstrate that visible-light irradiation of a mixture of alkyl carboxylic acids, [1.1.1]propellane, and bis(pinacolato)diboron (B2pin2) in the presence of DMSO enables alkylboration of [1.1.1]propellane. The reaction proceeds even in the absence of a catalyst, and the addition of an iron catalyst improves the yield (from 86% to 98%). The reaction conditions, substrate scope, and applications are investigated, and a mechanistic proposal is provided.

Indeed, the direct use of carboxylic acids as radical precursors for the alkylboration of [1.1.1]propellane appears synthetically useful.

However, for the mechanistic claims emphasized by the authors, the experimental evidence and mechanistic analysis are clearly insufficient. At present, the mechanistic claims do not match the data, and several key assumptions are not experimentally justified. Because these points lie at the very core of the manuscript, and significant gaps remain in the mechanistic support, I am unable to recommend this manuscript for publication in Nature Communications in its current form.

1. Lack of direct evidence for photoexcitation and homolysis of the B2pin2–DMSO complex

The authors claim that the B2pin2–DMSO complex absorbs 390-nm light and undergoes photolytic cleavage to produce the •OBpin radical. However, almost no direct evidence supporting this key assertion is provided. Although UV–vis spectra are shown, the molar absorptivity and detailed spectral features are missing, making it difficult to assess whether the complex absorbs light strongly enough to drive the reaction. The formation of DME or HOBpin is used to infer the intermediacy of Int-3 and the •OBpin radical (Int-4), but these products do not uniquely establish the proposed mechanism and could arise from alternative pathways. Critically important experiments—EPR spectroscopy, radical trapping, spin-trapping studies, HRMS detection of intermediates, and DFT calculations—are absent. As a result, the mechanistic hypothesis remains speculative and is not sufficiently supported to claim photolytic B–B bond cleavage. Furthermore, whether the •OBpin radical is capable of engaging in O–H HAT versus SET from carboxylic acids could be addressed by quantum chemical calculations, but such analysis is not provided.

2. The claim of a parallel LMCT pathway in the presence of iron is overstated

The authors state that HAT and LMCT proceed "simultaneously," yet the only supporting evidence for LMCT is the increased yield upon addition of Fe(III). No direct mechanistic data are shown. Considering photophysical competition, Fe(III) complexes typically have significantly stronger absorption in the ~390-nm region than a weakly absorbing B2pin2–DMSO complex. Therefore, it is entirely plausible that, under Fe-catalyzed conditions, photolysis of B2pin2–DMSO is negligible and LMCT is the dominant—perhaps exclusive—pathway. Based on the current data, the interpretation that either HAT or LMCT dominates depending on the presence or absence of iron is far more reasonable than the claim of two parallel pathways. To support an LMCT mechanism, Stern–Volmer quenching experiments, in-situ UV–vis analysis, or intermediate detection under photoexcitation of the Fe complex are needed. These measurements are essential for demonstrating that the ligand-to-metal charge-transfer pathway truly contributes to decarboxylation. Furthermore, since cesium carbonate appears to have a beneficial effect but is claimed not to participate in the mechanism, its role requires discussion and clarification.

Minor issues

3. The method of preparing [1.1.1]propellane is not described in the Supporting Information. Most commonly, propellane is generated using organolithium reagents, and depending on the preparation, the resulting solution may contain LiX salts or alkyl halides. Such species could influence either the absorption properties of the reaction mixture or the formation of

reactive intermediates. The authors should clearly state how propellane was prepared and whether impurities arising from the generation method could affect the observed reactivity.

4. On page 12, the phrase "serve an an O–H HAT reagent" contains a typographical error ("an an").

5. In Table 2, compound 3m is not bolded, unlike the other entries.

In summary, although the synthetic transformation is interesting and potentially useful, the mechanistic proposal is insufficiently substantiated and relies heavily on inference rather than direct evidence. Major additional experimental and/or computational mechanistic studies are required before the mechanistic claims can be considered convincing.

Version 1:

Reviewer comments:

Reviewer #1

(Remarks to the Author)

My comments and questions have been properly addressed, and I support the publication of the work in Nature Communications.

Reviewer #2

(Remarks to the Author)

The authors revised the manuscript according to the reviewers' comments. This reviewer confirmed that the authors have answered all of the questions and comments from the three reviewers by conducting experiments. The results obtained were incorporated into the revised manuscript, and the discussion therein successfully explained the core questions commonly raised by reviewers 2 and 3. Their faithful responses have improved the manuscript significantly. I believe the current version is almost acceptable. It is desirable to revise the following points (1, 3, 4) if possible.

1) Fig.1: Capital letters A-D were used to represent each subject. Lowercase letters are used in the main text and in the legend for Figure 1. Figure 1a should be changed to Figure 1A, for example.

2) Added examples of iron-free reactions revealed that B2pin2-DMSO alone is able to promote the target reaction. Although yields vary depending on the nature of the substrate, yields comparable to those of the iron-catalyzed reaction were obtained, suggesting that both reaction mechanisms may be involved in promoting the reaction.

3) Monitoring of the reaction using diphenyl sulfoxide and EPR measurement experiments reinforced the authors' ideas regarding the reaction mechanism. Normally, radical trapping experiments are combined with HRMS measurement to detect the adduct between a trapping agent and a radical intermediate. In this study, however, EPR measurements were used to infer the formation of Bpin-O radical by analogy referring to the spectra of similar compounds. I think the adduct of Bpin-O radical and DMPO (trapping agent) is very unstable and could not be detected by HRMS. Any comments?

4) Thanks to their response, I could understand that [1.1.1]propellane reacts with an alkyl radical so rapidly due to its large strain, while an alkyl radical reacts slowly with B2pin2. If this is the case, polymerization of BCP unit is prone to occur and therefore the use of 3 equivalents of B2pin2 seems essential. Any comments?

5) The chemoselective HAT of carboxylic acids by the Bpin-O radical is intriguing. It is likely that the formation of the complex between B2pin2, the acid, and DMSO is the key to this observed high selectivity. Alternatively, hydrogen bonding interactions between the acid and the oxygen atom of the boronate may be involved. I look forward to future research into this point.

Reviewer #3

(Remarks to the Author)

The authors sincerely addressed the reviewer's concern regarding the reaction mechanism by conducting additional experiments, which experimentally substantiated the proposed mechanism. Therefore, I consider that the manuscript can be accepted.

Version 2:

Reviewer comments:

Reviewer #2

(Remarks to the Author)

The authors sincerely responded to my additional comments and questions and all have been addressed properly. I enjoyed the discussion. I believe that this study would attract many chemists working in the related area, and therefore I strongly support the publication of the work in Nature Communications.

I found another correction in Figure 1C. According to my memory and reference 34, the reaction shown below was reported in 2025.

Reviewer #1 (Remarks to the Author):

In this manuscript, the authors present a straightforward and practical photoredox method for synthesizing BCP boronates from readily available aliphatic carboxylic acids. Under irradiation, a mixture of carboxylic acids, [1.1.1]propellane, and B2Pin2 in DMSO efficiently affords the target BCP boronates in good yields. Mechanistically, the authors propose a novel pathway involving photolytic cleavage of a B2Pin2–DMSO complex, which drives radical decarboxylation via hydrogen atom transfer. The addition of an iron catalyst further enhances the reaction yields by promoting decarboxylation through a ligand-to-metal charge transfer process. A wide range of carboxylic acids, including primary, secondary, and tertiary alkyl carboxylic acids are compatible with this system, without the need for prefunctionalization of the carboxylic acids. Thus, I recommend the publication of this work in Nature Communications after addressing the following comments.

We sincerely appreciate the reviewer's positive evaluation of our work and are grateful for this recognition. We also thank the reviewer for the detailed and constructive suggestions, which have significantly enhanced the clarity and completeness of the study.

1) Please comment on the scalability of this reaction system.

We thank the referee for this suggestion and agree that scalability is an important metric for new reactions. We have carried out our reaction on a larger 1 mmol scale and were delighted to isolate the desired product in 85% yield, suggesting the method to be amenable to larger scales. The details of our scaling up

experiments are shown below:

First, reaction was carried out following the general procedure in SI with 0.2 mmol scale. Carboxylic acid **2j** (56.9 mg, 0.4 mmol, 2.0 equiv.), bis(pinacolato)diboron (152.4 mg, 0.6 mmol, 3.0 equiv.), Cs₂CO₃ (13.0 mg, 0.04 mmol, 0.2 equiv.) and Fe(NO₃)₃·9H₂O (8.1 mg, 0.02 mmol, 0.1 equiv.) in DMSO (2.0 mL) under N₂ atmosphere was added [1.1.1]propellane **1** (0.23 mL, 0.2 mmol, 0.88 M in Et₂O, 1.0 equiv.). The isolated yield of the product **3j** is (51.1 mg, 0.176 mmol, 88%). Then reaction was carried out following the general procedure with 1.0 mmol scale while two 390 nm Kessil® lamps (25% intensity, 13 W) and a cooling fan were used. Carboxylic acid **2j** (284.4 mg, 2.0 mmol, 2.0 equiv.), bis(pinacolato)diboron (761.8 mg, 3.0 mmol, 3.0 equiv.), Cs₂CO₃ (65.2 mg, 0.2 mmol, 0.2 equiv.) and Fe(NO₃)₃·9H₂O (40.4 mg, 0.1 mmol, 0.1 equiv.) in DMSO (10.0 mL) under N₂ atmosphere was added [1.1.1]propellane **1** (1.1 mL, 1.0 mmol, 0.88 M in Et₂O,

1.0 equiv.). The isolated yield of the product **3j** is (246.7 mg, 0.85 mmol, 85%). Our experiments show that our borylation reaction can be easily scaled up to 1 mmol scale and the reaction setting up is as followings:

We thank the reviewer for encouraging us to assess scalability and for the valuable feedback that enabled us to gain deeper insight into the reaction properties.

2. The manuscript would be strengthened by a more detailed discussion on the limitations of the substrate scope and the functional group tolerance within the alkyl carboxylic acids. For the practical utility of this method, it is crucial to understand its compatibility with common functional groups. Key questions include: are free alcohols, amino groups, and heteroaromatics like pyridines tolerated?

We thank the referee for this insightful point and agree that more exploration of common functional groups is essential to understand the strengths and weaknesses of the reaction. Following the suggestion, we have performed further reaction scope exploration on substrates with a free alcohol, protected amino groups, and amino acid Boc-Pro-OH (substrate **3ao**, **3q** and **3aq** in the Table 2 of manuscript) and found these were tolerated under the standard reaction conditions. However, the chemistry does not apply to carboxylic acids containing pyridine groups. Additionally, the BCP boronate bearing aldehyde group **3ap** was successfully obtained in reasonable yield; however, this was via a two-step process involving oxidation of alcohol **3ao**.

We have added full characterization of the above compounds and referee can refer them to section 3.3 Characterization of MCR product in our revised supporting information.

3. Please carefully check the HMRS data in the supporting information, with particular attention to the calculated masses. For instance, regarding products 3a and 3b, the anticipated calculated mass values for 3a as [M+H]⁺ (C₁₆H₃₀BO₂⁺) and for 3b as [M+H]⁺ (C₁₅H₂₈BO₂⁺) should be 265.2333 and 251.2177, respectively.

We thank the referee for their careful attention to the HRMS data. The HRMS data in SI are correct and the mass facility who operated our samples used different software to calculate the calculated masses, which are slightly different from the calculated masses calculated by Chemdraw. please see the reports attached.

Target Compound Screening Report

Results Acquired by The University of Texas at Austin Mass Spectrometry Facility

Data File MSF25-0031(YW 208-3)_hrESIposLC1.d Sample Name 0031(YW 208-3) Comment 0031(YW 208-3)
 Position P1-E5 Instrument Name 6530 User Name
 Acq Method LC_C18_pos_jl.m Acquired Time 7/23/2025 5:53:11 AM DA Method MSF.m

MS Zoomed Spectrum

MS Spectrum Peak List

Obs. m/z	Calc. m/z	Charge	Abundance	Formula	Ion Species	Ygt Mass Error (ppm)
264.2365	264.2370	1	250452	C16H29BO2	(M+H)+	1.76
265.2334	265.2336	1	1085259	C16H29BO2	(M+H)+	0.94
266.2362	266.2369	1	194559	C16H29BO2	(M+H)+	2.71
267.2385	267.2397	1	20598	C16H29BO2	(M+H)+	4.75
268.2439	268.2424	1	1821	C16H29BO2	(M+H)+	-5.61

--- End Of Report ---

HRMS (ESI+) calc. for C₁₆H₂₉BO₂ [M+H]⁺265.2336 (report that we got), 265.2333 (ChemDraw), found 265.2334.

HRMS report of **3b**:

Target Compound Screening Report

Results Acquired by The University of Texas at Austin Mass Spectrometry Facility

Data File MSF25-0031(YW 205-2)_hrESIposLC1.d Sample Name 0031(YW 205-2) Comment 0031(YW 205-2)
Position P1-E2 Instrument Name 6530 User Name
Acq Method LC_C18_pos_jl.m Acquired Time 7/23/2025 4:44:09 AM DA Method MSF.m

MS Zoomed Spectrum

MS Spectrum Peak List

Obs. m/z	Calc. m/z	Charge	Abundance	Formula	Ion Species	Tgt Mass Error (ppm)
250.2209	250.2213	1	197741	C ₁₅ H ₂₇ BO ₂	(M+H) ⁺	1.65
251.2179	251.2180	1	877622	C ₁₅ H ₂₇ BO ₂	(M+H) ⁺	0.41
252.2206	252.2213	1	141449	C ₁₅ H ₂₇ BO ₂	(M+H) ⁺	2.58
253.2242	253.2240	1	16887	C ₁₅ H ₂₇ BO ₂	(M+H) ⁺	-0.57
255.2321	255.2294	1	9337	C ₁₅ H ₂₇ BO ₂	(M+H) ⁺	-10.49

--- End Of Report ---

Page 1 of 1

Printed at: 11:16 AM on: 7/23/2025

HRMS (ESI⁺) calc. for C₁₅H₂₇BO₂ [M+H]⁺+251.2180 (report that we got), 251.2177 (ChemDraw), found 251.2179.

We appreciate such important insight and have made changes to our manuscript, which we hope this response can address the reviewer's suggestions. We thank the reviewer for helping us improve the quality of this work!

Reviewer #2 (Remarks to the Author):

West et al. developed a new method for synthesizing BCP boronate esters. This reaction allows carboxylic acids to be used directly without converting them to redox-active esters, making it highly valuable in synthetic chemistry. Under iron catalyst conditions, the addition reaction to [1.1.1]-propellane proceeds smoothly, regardless of the carboxylic acid series, and the desired boronate esters can be obtained in good yields. Furthermore, tolerance for heterofunctional groups and aromatic rings has been confirmed.

We are grateful for the recognition from this referee for our work!

The reaction mechanism contains novel and interesting insight. It is intriguing that the reaction proceeds without the addition of an iron catalyst. It has been previously discussed that the (pin)BO oxy-radical is generated when a sulfoxide-B₂pin₂ complex is photoexcited (ref 37). Also, the detection of DMS and (pin)BOH in Fig. 3 experimentally supports the generation of the (pin)BO oxy-radical. However, it is puzzling that no molecules derived from the (pin)BO oxy-radical were detected in the first reaction of Fig. 3d.

We thank the reviewer for this question. We propose that the Bpin-OH was not detected by HRMS in the DMSO solution of B₂pin₂ due to lack of hydrogen atom source. Additionally, we also performed EPR experiments to further elucidate the formation of (pin)BO oxy-radical, with spin trapping experiments with DMPO suggesting the formation of the BPin-O• radical. The full details of our EPR study are shown below and included in our revised SI:

EPR study:

Spin-trap EPR experiment Photo spin-trapping reactions were set up as follows: in 1 ml DMSO which contained 5,5-dimethyl-1-pyrroline *N*-oxide (DMPO) (113.2 mg, 1.0 mmol, 10.0 equiv.), added (A) B₂pin₂ (76.2 mg, 0.3 mmol, 3.0 equiv.), (B) B₂pin₂ (76.2 mg, 0.3 mmol, 3.0 equiv.) plus acid **2j** (28.4 mg, 0.2 mmol, 2.0 equiv.), (C) B₂pin₂ (76.2 mg, 0.3 mmol, 3.0 equiv.) plus acid **2j** (28.4 mg, 0.2 mmol, 2.0 equiv.) and [1.1.1]propellane (0.1 mmol, 1.0 equiv.), and (D) B₂pin₂ (76.2 mg, 0.3 mmol, 3.0 equiv.) plus acid **2j** (28.4 mg, 0.2 mmol, 2.0 equiv.) and [1.1.1]propellane (0.1 mmol, 1.0 equiv.) and Fe(NO₃)₃·9 H₂O (4.0 mg, 0.01 mmol, 0.1 equiv.) and Cs₂CO₃ (6.5 mg, 0.02 mmol, 0.2 equiv.). The reaction mixtures were placed in 390

nm Kessil® light (25% intensity, 13 W) with a cooling fan stirring for 1 h before sampling with glass capillary tubes, which were then sealed with Critoseal and transferred to EPR cavity for measurements.

X-band EPR spectra of DMPO-radical adducts were recorded on a Bruker EMX spectrometer. EPR parameters were: frequency, 9.3 GHz; microwave power, 10 mW; scan range, 80 G; modulation frequency, 100 kHz; modulation amplitude, 1.0 G, time constant, 0.16 s, and receiver gain, 1×10^5 or 7.1×10^4 . One or four scans were acquired for each sample. The spectra were analyzed and simulated using WinEPR and SimFonia, respectively.

Results: significant amounts of radicals were captured using DMPO in the photoreactions of B_2pin_2 with DMSO solvent, both in the absence and presence of acid and iron catalysts (Fig. 1). On the other hand, no radical EPR signal was observed in the photoreaction in the absence of a spin-trapping reagent (Fig. 2), indicating that any radical intermediate produced in the photoreactions is transient. All the DMPO-radical adducts show a g value of 2.006. In the absence of acid or iron, the radical produced in the photoreaction with B_2pin_2 only is mainly an oxygen-centered radical as indicated by the hyperfine splitting constants, $a_N = 13.7$ G and $a_{\beta-H} = 11.7$ G, that are typical of DMPO adduct with an oxygen-centered radical (Figure 1A). The oxygen-centered radical most likely resides on the $\cdot OBpin$ moiety generated from the $B_2pin_2/DMSO$ complex. Moreover, the $a_N/a_{\beta-H}$ ratio of the DMPO-OB(OR)₂ adduct is 1.2, which is similar to those observed in DMPO-OOR adducts but noticeably larger than those of DMPO-OR adducts. The boron atom therefore affects the hyperfine splittings in the DMPO-OB(OR)₂ adduct in a similar pattern to the second oxygen atom in DMPO-OOR adducts.

EPR data of the acid-catalyzed photoreaction was more complicated (Figure 1B) compared to spectrum A. The extra EPR features suggest that DMPO captured both radicals- the oxygen-center O-Bpin radical (compared to spectrum A) and a carbon-center radical (compared to spectrum D, *vide infra*). Moreover, more EPR features were observed which may be due to different conformers of DMPO adduct(s) in the presence of acid. The radical captured in reactions C and D, catalyzed by both acid and **1** (C) plus iron and cesium carbonate (D), was mainly a carbon-centered radical, as indicated by the hyperfine splitting constants $a_N = 14.6$ G and $a_{\beta-H} = 21.3$ G and $a_N/a_{\beta-H} = 0.69$ that are typical of DMPO adducts with carbon-centered radicals, DMPO-CR (Figure 1D). We proposed that the oxygen centered radical observed in reaction A was generated from homolytic cleavage of B_2pin_2 -DMSO complex and the carbon-centered radical observed in reactions B – D was produced by decarboxylation of **2j**. It appears that the decarboxylation reaction was enhanced by addition of iron salt and base, as indicated by the significantly larger size of the EPR signal in spectrum D compared to the same signals in spectra B and C. Thus, we hypothesized that iron-mediated LMCT and $\cdot OBpin$ mediated HAT synergistically accelerated the decarboxylation process.

Fig.1. EPR spectra of the trapped radical species in the photoreactions. The EPR spectra of DMPO-trapped radical(s) in the photoreactions of (A) B₂pin₂ only, (B) B₂pin₂ in the presence of acid, (C) B₂pin₂ in the presence of acid **2j** and **1**, and (D) B₂pin₂ in the presence of acid **2j**, **1**, iron and cesium salts. Red dash lines: simulations of the major DMPO-radical adducts using the marked hyperfine splitting constants. The EPR data is normalized for receiver gain and number of scans for direct comparison. Spectra (A) – (C) are scaled by a factor of 5, marked by “x5” on the right, for easy visualization. The hyperfine splitting constants of nitrogen and β -proton of DMPO-radical adducts, a_N and $a_{\beta\text{-H}}$, are labeled for spectra (A) and (D).

Fig. 2. EPR spectrum of the photoreaction mixture in the absence of DMPO. The same reaction of (D) in the absence of DMPO. No EPR signal is observed.

We have also revised figure 3 and the surrounding main body text to incorporate these data as shown below:

Interestingly, replacing DMSO with different equivalents of diphenyl sulfoxide under iron and base free conditions still yielded the desired product **3j** from 32% to 46% yield, and diphenyl sulfide was also obtained in reasonable amounts (Fig. 3E, see SI for detailed information). No desired product **3j** was observed in the absence of sulfoxide source and this indicated sulfoxide is a key factor in the success of the reaction. EPR experiments were also carried out with addition of spin-trap reagent 5,5-dimethyl-1-pyrroline N-oxide (DMPO) (Fig. 3F). After irradiating a solution of B_2pin_2 in DMSO for 1 h, the solution was subjected to EPR analysis and, to our delighted, an oxygen-centered radical was observed as indicated by the hyperfine splitting constants, $a_N = 13.7$ G and $a_{\beta-H} = 11.7$ G, that are typical of DMPO adduct with an oxygen-centered radical (reaction a).³⁸ The oxygen-centered radical most likely originates from a B_2pin_2 -DMSO complex, suggesting the formation of $\cdot OBpin$ (see SI for EPR spectra and detailed results analysis). EPR analysis of a solution of carboxylic acid and B_2Pin_2 (reaction b) was more complicated, suggesting both oxygen centered and carbon centered radicals were captured by DMPO (see SI for EPR spectra and detailed results analysis). The radicals captured in reactions containing carboxylic acid, [1.1.1]propellane, and B_2Pin_2 in DMSO (reaction c) and the same mixture with catalytic iron (reaction d) were mainly carbon-centered radicals, as indicated by the hyperfine splitting constants $a_N = 14.6$ G and $a_{\beta-H} = 21.3$ G and $a_N/a_{\beta-H} = 0.69$ that are typical of DMPO adducts with carbon-centered radicals^{39,40} (see SI for EPR spectra and detailed

results analysis). These EPR data support the ability of B_2Pin_2 /sulfoxide mixtures to photolytically generate reactive oxygen species and achieve radical decarboxylation in the presence of carboxylic acids.

Comparison of carboxylic acid containing reactions supports that the decarboxylation is enhanced by addition of iron salt and base, as a significantly larger carbon-centered radical EPR signal for reaction d compared to the same signals for reaction b and c (see SI for EPR spectra and detailed results analysis). This enhanced decarboxylation is further supported by the higher yield of the desired product **3j** (98%) in the presence of iron catalyst. Taken together, these EPR and experimental results are consistent with an LMCT-mediated decarboxylation process contributing productively to the reaction.

Fig. 3. (A) TEMPO involved radical scavenger experiment. (B) Radical clock experiment. (C) UV-vis spectra. (D) HRMS detection results. (E) DMSO replacement experiments. (F) EPR experiments. (G) Proposed reaction mechanism.

The mechanistic proposal is made based on the detection of DMS and pinBOH, but no quantitative discussion is provided. Considering that the diboron has a reducing nature and DMSO is oxidative, it is insufficient to propose Fig. 3e for the HAT cycle based on their detection alone. If Ph₂S=O could be used instead of DMSO, quantitative discussion would be possible..

We thank the reviewer for giving us this suggestion and carried out the suggested experiments to allow a more quantitative discussion. When replacing DMSO with the suggested diphenylsulfoxide, the desired product **3j** was successfully obtained in yields ranging from 32% to 46% and the corresponding diphenyl sulfide was also formed under iron and base free conditions. No desired product **3j** was observed in the absence of sulfoxide source under iron and base free conditions and in all product-forming cases a superstoichiometric amount of diphenyl sulfide was formed compared to product **3j**, confirming that it participates in the reaction in reagent quantities. We believe these results further support the necessity of sulfoxide for product formation and its role as an oxygen atom donor given the formation of diphenyl sulfide for the success of the reactions. The details of quantitative discussion are shown below:

Reactions were carried out following the general procedure with 0.1 mmol scale in the absence of iron and base when DMSO was replaced with different equivalents of solid diphenyl sulfoxide.

Entry	Equivalence of phenyl sulfone	solvent	Yield of 3j	Amount of 4j formed
1	0	MeCN	0	0 mmol
2	5.0	MeCN	32%	0.053 mmol
3	5.0	DMA	11%	0.033 mmol
4	5.0	DMF	7%	0.019 mmol
5	10.0	MeCN	41%	0.067 mmol
6	15.0	MeCN	46%	0.087 mmol

NMR yield in each entry was obtained by adding 16 μ L DCE (0.2 mmol) as internal standard.
Entry 1: no desired product **3j** formed in the absence of sulfoxide

TLC (PMA stain) shows no desired product 3j observed without addition of phenyl sulfoxide.

Entry 2: 4.14 (DCE, 4H):1.00 ([1.1.1]propellane, 6H), (4.14/4) : (1/6) = 1.035:0.167
0.2 mmol/(1.035/0.167) = 0.032 mmol, yield of 3j: 0.032 mmol/0.1 mmol = 32%

Amount of 4j formed: $(4.14/4) : (2.75/10) = 1.035:0.275$, $0.2 \text{ mmol}/(1.035/0.275) = 0.053 \text{ mmol}$

Entry 3: 12.01 (DCE, 4H):1.00 ([1.1.1]propellane, 6H), $(12.01/4) : (1/6) = 3.0025:0.167$
 $0.2 \text{ mmol}/(3.0025/0.167) = 0.011 \text{ mmol}$, yield of 3j: $0.011 \text{ mmol}/0.1 \text{ mmol} = 11\%$
Amount of 4j formed: $(12.01/4) : (5.04/10) = 3.0025:0.504$, $0.2 \text{ mmol}/(3.0025/0.504) = 0.033 \text{ mmol}$

Entry 4: 19.79 (DCE, 4H):1.00 ([1.1.1]propellane, 6H), $(19.79/4) : (1/6) = 4.9475:0.167$

0.2 mmol/(4.9475/0.167) = 0.0068 mmol, yield of 3j: 0.0068 mmol/0.1 mmol = 7%
 Amount of 4j formed:(19.79/4) : (4.80/10) = 4.9475:0.480, 0.2 mmol/(4.9475/0.480) = 0.019 mmol

Entry 5: 3.22 (DCE, 4H):1.00 ([1.1.1]propellane, 6H), (3.22/4) : (1/6) =0.805:0.167
 0.2 mmol/(0.805/0.167) = 0.041 mmol, yield of 3j: 0.041 mmol/0.1 mmol = 41%
 Amount of 4j formed:(3.22/4) : (2.68/10) = 0.805:0.268, 0.2 mmol/(0.805/0.268) = 0.067 mmol

Entry 6: 2.89 (DCE, 4H):1.00 ([1.1.1]propellane, 6H), (2.89/4) : (1/6) =0.7225:0.167

$0.2 \text{ mmol}/(0.7225/0.167) = 0.046 \text{ mmol}$, yield of 3j: $0.046 \text{ mmol}/0.1 \text{ mmol} = 46\%$
Amount of 4j formed: $(2.89/4) : (3.13/10) = 0.7225:0.313$, $0.2 \text{ mmol}/(0.7225/0.313) = 0.087 \text{ mmol}$

The DMSO replacement experiments demonstrate that sulfoxide (DMSO and phenyl sulfoxide) is a key factor for the successful reactions under both iron and base-free conditions. Based on the above experiments, we hypothesize that B_2pin_2 may have coordinated with DMSO and released OBpin radical.

Furthermore, the fact that the carbon radical derived from the carboxylic acid preferentially adds to BCP and not to diboron has not been well elucidated. It is desirable to provide an explanation, citing references as necessary

This is an insightful question. We believe the carbon centered radical derived from the carboxylic acid adds preferentially to [1.1.1]propellane for several reasons, including that the strain release-driven radical attack on [1.1.1]propellane is known to occur exceptionally rapidly and addition to B_2Pin_2 is known to be relatively slow. In addition, the nucleophilic alkyl radical derived from decarboxylation of the successful carboxylic acids in our reaction exhibits a polarity mismatch with B_2pin_2 , further contributing to slow addition. Taken together, these factors can help explain why we exclusively form BCP boronate products and do not observe direct decarboxylative borylation products.

Reference 11: Fawcett, A.; Pradeilles, J.; Wang, Y.; Mutsuga, T.; Myers, E. L.; Aggarwal, V. K. Photoinduced Decarboxylative Borylation of Carboxylic Acids. *Science* 357, 283-286 (2017) [DOI: 10.1126/science.aan3679](https://doi.org/10.1126/science.aan3679)

and reference 14: Dong, W.; Yen-Pon, E.; Li, L.; Bhattacharjee, A.; Jolit, A.; Molander, G. A. Exploiting the sp^2 Character of Bicyclo[1.1.1]Pentyl Radicals in the Transition-Metal-Free Multi-Component Difunctionalization of [1.1.1]Propellane. *Nat. Chem.* 14, 1068–1077 (2022). <https://doi.org/10.1038/s41557-022-00979-0> were cited in the manuscript.

We appreciate the reviewer's perspective on this point!

A novel aspect is that the (pin)BO oxy-radical causes HAT selectively for carboxylic acid O-H bond, but unfortunately there is no discussion for the reason. The starting carboxylic acid and the product contain C-H bonds with significantly smaller BDEs than the carboxylic acid O-H bond, and yet high selectivity for carboxylic acid and (estimated) BDE value of (pin)BO-H bond are of great interest. The changes in the UV-Vis absorption spectrum (Fig. 3c) strongly suggest an interaction between the carboxylic acid and the diboron, which might provide a clue to explaining the chemoselectivity. A clear explanation for this, based on

experimental or theoretical data, is necessary for publication in Nature Commun which requires a high level of academic merit.

We thank referee for raising this important point and have pursued additional experiments to further explore this chemoselectivity by including a range of potential substrates with weaker C–H bonds.

Different equivalents of methylcyclohexane (5 equiv./10 equiv./ 20 equiv.), toluene (5 equiv./10 equiv./ 20 equiv.) and *p*-anisaldehyde (5 equiv./10 equiv./ 20 equiv.) instead of carboxylic acid were tested under standard reaction conditions in the absence of iron and base but none of the desired BCP boronates were detected. At the same time, we have significant evidence from both HRMS and EPR studies that •OBPin formation does not require carboxylic acid, suggesting that •OBPin should still be formed in all these cases.

These results suggest that •OBpin radical cannot easily seize the H from the C-H bonds of these compounds despite each having C-H bonds with significantly smaller BDEs than the carboxylic acid O-H bond. As the reviewer helpfully highlighted, the UV-Vis study is suggestive of association between B₂Pin₂ and carboxylic acid and we speculate that coordination of the carboxylic acid with B₂pin₂, may make the O-H bond more vulnerable to attack by OBpin radical.

In this context, it is debatable that their unique HAT cycle applies only to carboxylic acid 2j and the generality of the catalytic system has not been investigated at all.

We agree with the referee that the generality of this HAT Cycle system should be further investigated to explore whether it is limited to 2j. We explored the suitability of the HAT Cycle for **primary carboxylic acid 3e** (23%), **secondary carboxylic acid 3h** (61%), **tertiary carboxylic acid 3k** (64%), **benzylic carboxylic acid 3x** (35%) and **drug 3al** (70%) and confirmed that all rendered us the desired product with reasonable isolated yields, suggesting this system to be relatively general for product formation.

This study reports a synthetically useful reaction and thus I am positive about publication in this journal. However, there are many ambiguities regarding the novel HAT cycle, which I think requires extensive revisions. Other minor errors that I have noticed are as follows:

Abstract: line 15: direct synthesis of BCP. “of” is missing.

We thank reviewer for drawing our attention to this error! We have added the missing “of” to the abstract.

Page 2: line 49: N-hydroxyphthalimide. “H” should be lowercase.

We thank reviewer for this point! We have corrected “H” to lowercase.

P4, line 70: 1,3-disubstituted. “di” is missing.

We thank reviewer for pointing out the mistake and we have added “di” in our manuscript.

P13, line 226: “the proton of a carboxylic acid” should be corrected to “the hydrogen atom of a carboxylic acid O-H.”

We thank reviewer for raising this important point! We have corrected it to “the hydrogen atom of a carboxylic acid O-H” in our manuscript.

P14, Fig 3c: Legends are too small. Please use the picture with much higher resolution.

We thank reviewer for the suggestion. We have revised fig 3c in manuscript to have much more readable legends and we also added the very clear illustration of these results on page 43 of the SI.

We hope our detailed revision has adequately addressed the points raised by this reviewer and we extremely grateful for the referee’s recognition to our work. These suggestions have helped us improve the quality of our manuscript!

Reviewer #3 (Remarks to the Author):

West and co-workers report a new synthetic approach to bicyclo[1.1.1]pentane (BCP). BCP has recently attracted significant attention as a bioisostere of para-substituted benzene, and two main strategies have been widely explored: those using [1.1.1]propellane and those using bicyclo[1.1.0]butane as precursors. The present manuscript falls into the former category. The authors demonstrate that visible-light irradiation of a mixture of alkyl carboxylic acids, [1.1.1]propellane, and bis(pinacolato)diboron (B2pin2) in the presence of DMSO enables alkylboration of [1.1.1]propellane. The reaction proceeds even in the absence of a catalyst, and the addition of an iron catalyst improves the yield (from 86% to 98%). The reaction conditions, substrate scope, and applications are investigated, and a mechanistic proposal is provided.

Indeed, the direct use of carboxylic acids as radical precursors for the alkylboration of [1.1.1]propellane appears synthetically useful. However, for the mechanistic claims emphasized by the authors, the experimental evidence and mechanistic analysis are clearly insufficient. At present, the mechanistic claims do not match the data, and several key assumptions are not experimentally justified. Because these points lie at the very core of the manuscript, and significant gaps remain in the mechanistic support, I am unable to recommend this manuscript for publication in Nature Communications in its current form.

We appreciate the reviewers’ recognition of the potential merits of our work and have used the feedback of all three referees to further support our hypotheses and reveal additional insights into the mechanism.

1. Lack of direct evidence for photoexcitation and homolysis of the B2pin2–DMSO complex

The authors claim that the B2pin2–DMSO complex absorbs 390-nm light and undergoes photolytic cleavage to produce the $\cdot\text{OBpin}$ radical. However, almost no direct evidence supporting this key assertion is provided. Although UV–vis spectra are shown, the molar absorptivity and detailed spectral features are missing, making it difficult to assess whether the complex absorbs light strongly enough to drive the reaction. The formation of DME or HOBpin is used to infer the intermediacy of Int-3 and the $\cdot\text{OBpin}$ radical (Int-4), but these products do not uniquely establish the proposed mechanism and could arise from alternative pathways. Critically important experiments—EPR spectroscopy, radical trapping, spin-trapping studies, HRMS detection of intermediates, and DFT calculations—are absent. As a result, the mechanistic hypothesis remains speculative and is not sufficiently supported to claim photolytic B–B bond cleavage. Furthermore, whether the $\cdot\text{OBpin}$ radical is capable of engaging in O–H HAT versus SET from carboxylic acids could be addressed by quantum chemical calculations, but such analysis is not provided.

We thank the referee for raising these important points and we have pursued a number of additional experiments to explore and more strongly support this mechanistic proposal. First, to elucidate the generation of $\cdot\text{OBpin}$ radical, we performed a series of EPR experiments to detect the $\cdot\text{OBpin}$ radical as shown below, summarized in Figure 3F, and included in the revised SI:

EPR study:

Spin-trap EPR experiment Photo spin-trapping reactions were set up as follows: in 1 ml DMSO which contained 5,5-dimethyl-1-pyrroline *N*-oxide (DMPO) (113.2 mg, 1.0 mmol, 10.0 equiv.), added (A) B₂pin₂ (76.2 mg, 0.3 mmol, 3.0 equiv.), (B) B₂pin₂ (76.2 mg, 0.3 mmol, 3.0 equiv.) plus acid **2j** (28.4 mg, 0.2 mmol, 2.0 equiv.), (C) B₂pin₂ (76.2 mg, 0.3 mmol, 3.0 equiv.) plus acid **2j** (28.4 mg, 0.2 mmol, 2.0 equiv.) and [1.1.1]propellane (0.1 mmol, 1.0 equiv.), and (D) B₂pin₂ (76.2 mg, 0.3 mmol, 3.0 equiv.) plus acid **2j** (28.4 mg, 0.2 mmol, 2.0 equiv.) and [1.1.1]propellane (0.1 mmol, 1.0 equiv.) and Fe(NO₃)₃·9 H₂O (4.0 mg, 0.01 mmol, 0.1 equiv.) and Cs₂CO₃ (6.5 mg, 0.02 mmol, 0.2 equiv.). The reaction mixtures were placed in 390 nm Kessil® light (25% intensity, 13 W) with a cooling fan stirring for 1 h before sampling with glass capillary tubes, which were then sealed with Critoseal and transferred to EPR cavity for measurements.

X-band EPR spectra of DMPO-radical adducts were recorded on a Bruker EMX spectrometer. EPR parameters were: frequency, 9.3 GHz; microwave power, 10 mW; scan range, 80 G; modulation frequency, 100 kHz; modulation amplitude, 1.0 G, time constant, 0.16 s, and receiver gain, 1 × 10⁵ or 7.1 × 10⁴. One or four scans were acquired for each sample. The spectra were analyzed and simulated using WinEPR and SimFonia, respectively.

Results: significant amounts of radicals were captured using DMPO in the photoreactions of B₂pin₂ with DMSO solvent, both in the absence and presence of acid and iron catalysts (Fig. 1). On the other hand, no radical EPR signal was observed in the photoreaction in the absence of a spin-trapping reagent (Fig. 2), indicating that any radical intermediate produced in the photoreactions is transient. All the DMPO-radical adducts show a *g* value of 2.006. In the absence of acid or iron, the radical produced in the photoreaction with B₂pin₂ only is mainly an oxygen-centered radical as indicated by the hyperfine splitting constants, *a_N* = 13.7 G and *a_{β-H}* = 11.7 G, that are typical of DMPO adduct with an oxygen-centered radical (Figure 1A). The oxygen-centered radical most likely resides on the •OBpin moiety generated from the B₂pin₂/DMSO complex. Moreover, the *a_N*/*a_{β-H}* ratio of the DMPO-OB(OR)₂ adduct is 1.2, which is similar to those observed in DMPO-OOR adducts but noticeably larger than those of DMPO-OR adducts. The boron atom therefore affects the hyperfine splittings in the DMPO-OB(OR)₂ adduct in a similar pattern to the second oxygen atom in DMPO-OOR adducts.

EPR data of the acid-catalyzed photoreaction was more complicated (Figure 1B) compared to spectrum A. The extra EPR features suggest that DMPO captured both radicals- the oxygen-center O-Bpin radical (compared to spectrum A) and a carbon-center radical (compared to spectrum D, *vide infra*). Moreover, more EPR features were observed which may be due to different conformers of DMPO adduct(s) in the

presence of acid. The radical captured in reactions C and D, catalyzed by both acid and **1** (C) plus iron and cesium carbonate (D), was mainly a carbon-centered radical, as indicated by the hyperfine splitting constants $a_N = 14.6$ G and $a_{\beta-H} = 21.3$ G and $a_N/a_{\beta-H} = 0.69$ that are typical of DMPO adducts with carbon-centered radicals, DMPO-CR (Figure 1D). We proposed that the oxygen centered radical observed in reaction A was generated from homolytic cleavage of B₂pin₂-DMSO complex and the carbon-centered radical observed in reactions B – D was produced by decarboxylation of **2j**. It appears that the decarboxylation reaction was enhanced by addition of iron salt and base, as indicated by the significantly larger size of the EPR signal in spectrum D compared to the same signals in spectra B and C. Thus, we hypothesized that iron-mediated LMCT and •OBpin mediated HAT synergistically accelerated the decarboxylation process.

Fig.1. EPR spectra of the trapped radical species in the photoreactions. The EPR spectra of DMPO-trapped radical(s) in the photoreactions of (A) B_2pin_2 only, (B) B_2pin_2 in the presence of acid, (C) B_2pin_2 in the presence of acid **2j** and **1**, and (D) B_2pin_2 in the presence of acid **2j**, **1**, iron and cesium salts. Red dash lines: simulations of the major DMPO-radical adducts using the marked hyperfine splitting constants. The EPR data is normalized for receiver gain and number of scans for direct comparison. Spectra (A) – (C) are scaled by a factor of 5, marked by “x5” on the right, for easy visualization. The hyperfine splitting constants of nitrogen and β -proton of DMPO-radical adducts, a_N and $a_{\beta\text{-H}}$, are labeled for spectra (A) and (D).

Fig. 2. EPR spectrum of the photoreaction mixture in the absence of DMPO. The same reaction of (D) in the absence of DMPO. No EPR signal is observed.

These data provide much stronger support for the intermediacy of the $\cdot\text{OBPin}$ radical and its formation from the photolysis of a $\text{B}_2\text{Pin}_2/\text{DMSO}$ mixture. These data are complemented by insightful experiments we pursued at the suggestion of reviewer 2, where we replaced DMSO solvent with reagent quantities of diphenyl sulfoxide as shown below and included in the revised SI:

Reactions were carried out following the general procedure with 0.1 mmol scale in the absence of iron and base when DMSO was replaced with different equivalents of solid diphenyl sulfoxide.

Entry	Equivalence of phenyl sulfone	solvent	Yield of 3j	Amount of 4j formed
1	0	MeCN	0	0 mmol
2	5.0	MeCN	32%	0.053 mmol

3	5.0	DMA	11%	0.033 mmol
4	5.0	DMF	7%	0.019 mmol
5	10.0	MeCN	41%	0.067 mmol
6	15.0	MeCN	46%	0.087 mmol

NMR yield in each entry was obtained by adding 16 μ L DCE (0.2 mmol) as internal standard.
Entry 1: no desired product 3j formed in the absence of sulfoxide

TLC (PMA stain) shows no desired product 3j observed without addition of phenyl sulfoxide.

Entry 2: 4.14 (DCE, 4H):1.00 ([1.1.1]propellane, 6H), (4.14/4) : (1/6) = 1.035:0.167
0.2 mmol/(1.035/0.167) = 0.032 mmol, yield of 3j: 0.032 mmol/0.1 mmol = 32%
Amount of 4j formed:(4.14/4) : (2.75/10) = 1.035:0.275, 0.2 mmol/(1.035/0.275) = 0.053 mmol

Entry 3: 12.01 (DCE, 4H):1.00 ([1.1.1]propellane, 6H), (12.01/4) : (1/6) =3.0025:0.167
 0.2 mmol/(3.0025/0.167) = 0.011 mmol, yield of 3j: 0.011 mmol/0.1 mmol = 11%
 Amount of 4j formed:(12.01/4) : (5.04/10) = 3.0025:0.504, 0.2 mmol/(3.0025/0.504) = 0.033 mmol

Entry 4: 19.79 (DCE, 4H):1.00 ([1.1.1]propellane, 6H), (19.79/4) : (1/6) =4.9475:0.167
 0.2 mmol/(4.9475/0.167) = 0.0068 mmol, yield of 3j: 0.0068 mmol/0.1 mmol = 7%
 Amount of 4j formed:(19.79/4) : (4.80/10) = 4.9475:0.480, 0.2 mmol/(4.9475/0.480) = 0.019 mmol

Entry 5: 3.22 (DCE, 4H):1.00 ([1.1.1]propellane, 6H), (3.22/4) : (1/6) =0.805:0.167
0.2 mmol/(0.805/0.167) = 0.041 mmol, yield of 3j: 0.041 mmol/0.1 mmol = 41%
Amount of 4j formed:(3.22/4) : (2.68/10) = 0.805:0.268, 0.2 mmol/(0.805/0.268) = 0.067 mmol

Entry 6: 2.89 (DCE, 4H):1.00 ([1.1.1]propellane, 6H), (2.89/4) : (1/6) =0.7225:0.167
0.2 mmol/(0.7225/0.167) = 0.046 mmol, yield of 3j: 0.046 mmol/0.1 mmol = 46%
Amount of 4j formed:(2.89/4) : (3.13/10) = 0.7225:0.313, 0.2 mmol/(0.7225/0.313) = 0.087 mmol

These DMSO replacement experiments demonstrate that sulfoxide (DMSO or phenyl sulfoxide) is a key factor for the successful reactions under both iron and base-free conditions. Further, product formation is directly coupled to conversion of sulfoxide to sulfide, demonstrating that reduction of sulfoxide is involved in the product-forming process. These data are further corroborated by HRMS data of various control reactions showing the formation of both sulfide and HOBPin (the successor product to \cdot OBPin) in combinations containing B_2Pin_2 , DMSO, and carboxylic acid shown below and in figure 3D:

The positive detection of HOBPin in carboxylic acid-containing mixtures is consistent with its formation via HAT-mediated decarboxylation. This is further supported by the failure to detect HOBPin in the absence of carboxylic acid, suggesting \cdot OBPin to be particularly adept at this HAT (or formal HAT) process.

To gain additional insight into the importance of this homolysis process for

Therefore, we hypothesize that \cdot OBpin radicals were generated based on EPR results. We also think the \cdot OBpin radical is capable of engaging in O–H HAT versus SET from carboxylic acids as there is no carboxylate generation in our standard reaction conditions in the absence of iron and base (shown below) and it's unlikely to go through SET process.

We appreciate the reviewer prompting us to do more studies to strengthen our proposed mechanism! We have also added this part of study to the section '3.5 Mechanism study' in the revised supporting information.

2. The claim of a parallel LMCT pathway in the presence of iron is overstated

The authors state that HAT and LMCT proceed "simultaneously," yet the only supporting evidence for LMCT is the increased yield upon addition of Fe(III). No direct mechanistic data are shown. Considering photophysical competition, Fe(III) complexes typically have significantly stronger absorption in the ~390-nm region than a weakly absorbing B2pin2–DMSO complex. Therefore, it is entirely plausible that, under Fe-catalyzed conditions, photolysis of B2pin2–DMSO is negligible and LMCT is the dominant—perhaps exclusive—pathway. Based on the current data, the interpretation that either HAT or LMCT dominates depending on the presence or absence of iron is far more reasonable than the claim of two parallel pathways.

To support an LMCT mechanism, Stern–Volmer quenching experiments, in-situ UV–vis analysis, or intermediate detection under photoexcitation of the Fe complex are needed. These measurements are essential for demonstrating that the ligand-to-metal charge-transfer pathway truly contributes to decarboxylation.

We thank reviewer for the question and agree that several mechanistic possibilities might be plausible. Considering all data we have obtained over the course of this project and revision, we have proposed the proposed the parallel HAT and LMCT mechanism based on the following observations:

First, as the reviewer mentions in their response, we observed higher, albeit not dramatically higher, yields of the desired products in the presence of catalytic iron. As an example, we obtained high yield of **3j** (83%, 86%) under the iron-free conditions and a higher (~15%) yield of the desired product **3j** (98%, table 1 in manuscript) in the presence of iron. As the reaction in the absence of iron has been demonstrated to be highly competent on the same timescale as the iron reaction for product formation, we believe it is likely that this process continues to function even in the presence of iron.

Secondly, we performed intermediate detection experiments of both the iron containing and iron free reactions via EPR as described above and observed that the decarboxylation reaction appeared to be enhanced by addition of both iron salt and base as indicated by the significantly larger size of the carbon-centered radical EPR signal in spectrum D (HAT+LMCT cycle) compared to the same signal in C (pure "HAT cycle", or iron-free conditions) (shown above in answer to question 1).

Third, we obtained significant reactivity support for an LMCT process when MeCN was used instead of DMSO under standard conditions with iron and base, where the desired product **3j** was still obtained with 31% yield (entry 6, table 1 in manuscript). These conditions are similar to many published LMCT reaction condition (iron and base in MeCN), providing circumstantial support for this reactivity mode. This support is significantly strengthened by the observation that no desired product was detected when DMSO was replaced with MeCN under iron and base free conditions, indicating iron and base to be critical for reactivity in these sulfoxide-free conditions. We acknowledge that the change to MeCN might also perturb the underlying mechanism; however, the fact that the same product is formed in both cases and the other components of the reaction are unchanged suggests that a similar process is occurring. Similar mechanistic behavior is further supported by the recovery of iron-free reactivity in MeCN when diphenyl sulfoxide is added in reagent quantities as described above.

Taking these observations together, we observe a clear beneficial effect to both carbon-centered radical formation and product formation in the presence of iron; however, the enhancement in both cases is not so dramatic as to suggest a complete change in mechanism, making us uncomfortable to completely discount the iron-independent reaction in the presence of iron. Toward obtaining positive evidence for an iron-mediated photodecarboxylation process, we demonstrated that BCP boronate product can still be formed in the absence of sulfoxide when catalytic iron is present and no product is formed in the absence of iron and sulfoxide. These data in aggregate appear most consistent with the possibility of both iron-dependent and iron-independent reactions being present in our standard conditions.

Furthermore, since cesium carbonate appears to have a beneficial effect but is claimed not to participate in the mechanism, its role requires discussion and clarification.

We thank the reviewer for this opportunity to improve the completeness of our mechanistic proposal. Cesium carbonate appears to have a beneficial effect because it can deprotonate the carboxylic acid, thereby enabling coordination between carboxylate and iron for LMCT homolysis. We have modified our mechanism and included it to our proposed mechanism.

Minor issues

3. The method of preparing [1.1.1]propellane is not described in the Supporting Information. Most commonly, propellane is generated using organolithium reagents, and depending on the preparation, the resulting solution may contain LiX salts or alkyl halides. Such species could influence either the absorption properties of the reaction mixture or the formation of reactive intermediates. The authors should clearly state how propellane was prepared and whether impurities arising from the generation method could affect the observed reactivity.

We thank the reviewer for raising this concern. As suggested, we have added our procedure for [1.1.1]propellane preparation to the revised SI as shown below:

According to a literature procedure (Pickford, H. D.; Nugent, J.; Owen, B.; Mousseau, James. J.; Smith, R. C.; Anderson, E. A. Twofold Radical-Based Synthesis of N,C-Difunctionalized Bicyclo[1.1.1]pentanes. J. Am. Chem. Soc. 143, 9729-9736 (2021)).

PhLi (1.9 M in Bu₂O, 100 mL, 190 mmol) was added to a solution of 1,1-dibromo-2,2-bis(chloromethyl)-cyclopropane (28.2 g, 95.0 mmol) in Et₂O (60 mL, 3.2 M) in a 500 mL flask at – 40 °C. The mixture was then stirred at 0 °C for 2 h to give a yellow/brown suspension. A rotary evaporator was cleaned with acetone and Et₂O, then dried under vacuum and flushed with N₂. The reaction mixture was distilled under reduced pressure using the rotary evaporator, maintaining a steady drip of [1.1.1]propellane solution (water bath 25 °C, condenser and trap – 78 °C, 200 mbar → 50 mbar). Once the distillation was complete, the system was flushed with N₂ and the distilled solution was transferred to a flame dried AcroSeal™ bottle with septum under N₂ at – 78 °C. The remaining residue was quenched with MeOH/acetone. Total Volume = 58.0 mL (0.88 M in Et₂O), 54% yield. Another bath of [1.1.1]propellane was also prepared following the above procedure. Total Volume = 65 mL (0.69 M in Et₂O), 47% yield. ¹H NMR (400 MHz, CDCl₃) δ 1.94 (6H, s). Spectroscopic data in agreement with that reported previously.

Concentration was determined by ¹H NMR spectroscopy using a sample of stock solution (200 μL), DCE (50 μL) and CDCl₃ in NMR tube.

Calculation of concentration of [1.1.1]propellane batch 1:

2.40 (DCE, 4H) : 1.00 ([1.1.1]propellane, 6H)

(2.40/4) : (1.00/6) = 0.6 : 0.167

(50 μL x 1.253 g/mL)/98.96 g/mol = 0.63 mmol

$$(0.63 \text{ mmol}/(0.6/0.167))/0.2 \text{ mL} = 0.88 \text{ M}$$

Calculation of concentration of [1.1.1]propellane **batch 2**:

$$3.07 \text{ (DCE, 4H)} : 1.00 \text{ ([1.1.1]propellane, 6H)}$$

$$(3.07/4) : (1.00/6) = 0.7675 : 0.167$$

$$(50 \mu\text{L} \times 1.253 \text{ g/mL})/98.96 \text{ g/mol} = 0.63 \text{ mmol}$$

$$(0.63 \text{ mmol}/(0.7675/0.167))/0.2 \text{ mL} = 0.69 \text{ M}$$

Both [1.1.1]propellane from batch 1 and batch 2 were tested following the general procedure with carboxylic acid **2j** and rendered the desired product **3j** in 90% and 88% isolated yield respectively. Both Et₂O and Bu₂O do not affect the reactions.

4. On page 12, the phrase “serve an an O–H HAT reagent” contains a typographical error (“an an”).

We thank referee for pointing out the mistake and we have corrected the extra “an” to “as” in our manuscript.

5. In Table 2, compound **3m** is not bolded, unlike the other entries.

We thank referee for noting this error, which has now been corrected in our manuscript.

We are grateful for the insightful suggestions from all the referees and genuinely appreciate the editor’s and reviewers’ dedication to help us improve the quality of our work! We strongly believe the completeness and scholarship of our manuscript has been improved via this constructive process!

Reviewer #1 (Remarks to the Author):

My comments and questions have been properly addressed, and I support the publication of the work in Nature Communications.

We thank the reviewer for their thoughtful comments and constructive engagement with our work! Their feedback has been invaluable for improving the clarity and completeness of our study.

Reviewer #2 (Remarks to the Author):

The authors revised the manuscript according to the reviewers' comments. This reviewer confirmed that the authors have answered all of the questions and comments from the three reviewers by conducting experiments. The results obtained were incorporated into the revised manuscript, and the discussion therein successfully explained the core questions commonly raised by reviewers 2 and 3. Their faithful responses have improved the manuscript significantly. I believe the current version is almost acceptable. It is desirable to revise the following points (1, 3, 4) if possible.

We sincerely thank the reviewer for their continued dedication to improving our study! Their constructive comments have been invaluable to help us to gain further insight into the underlying mechanism and clarify the presentation of our results.

1) Fig.1: Capital letters A-D were used to represent each subject. Lowercase letters are used in the main text and in the legend for Figure 1. Figure 1a should be changed to Figure 1A, for example.

We thank the reviewer for bringing this inconsistency to your attention! We have corrected all instances of Fig. 1a to Fig. 1A, Fig. 1b to Fig. 1B, Fig. 1c to Fig. 1C and Fig. 1d to Fig. 1D.

2) Added examples of iron-free reactions revealed that B₂pin₂-DMSO alone is able to promote the target reaction. Although yields vary depending on the nature of the substrate, yields comparable to those of the iron-catalyzed reaction were obtained, suggesting that both reaction mechanisms may be involved in promoting the reaction.

We sincerely thank the reviewer for this insightful comment. We agree that the additional iron-free examples much more strongly demonstrate the ability of B₂pin₂-DMSO alone to promote the reaction, with comparable yields possible depending on the substrate. Similarly, these results give stronger support for the possible involvement of both reaction pathways in facilitating the transformation.

3) Monitoring of the reaction using diphenyl sulfoxide and EPR measurement experiments reinforced the authors' ideas regarding the reaction mechanism. Normally, radical trapping experiments are combined with HRMS measurement to detect the adduct between a trapping agent and a radical intermediate. In this study, however, EPR measurements were used to infer the formation of Bpin-O radical by analogy referring to the spectra of similar compounds. I think the adduct of Bpin-O radical and DMPO (trapping agent) is very unstable and could not be detected by HRMS. Any comments?

We thank reviewer for this point! We have not measured the lifetime of the BPin-O radical and DMPO; however, we anticipate it to be relatively short based on related compounds. The half-life of the DMPO-OH adduct is ~55 min and the DMPO adduct of a carbon-centered radical is ~>1 hr at neutral pH. While these lifetimes are significant, the half-life of the related DMPO-OOH adduct is significantly shorter than that of DMPO-OH (see *J. Phys. Chem. A* 2003, 107, 4407–4414. <https://doi.org/10.1021/jp027829f> and *Free Radic. Biol. Med.* 2003, 34, 1473-1481. [https://doi.org/10.1016/S0891-5849\(03\)00182-5](https://doi.org/10.1016/S0891-5849(03)00182-5)), suggesting that substitution at the oxygen atom of DMPO adducts can change the stability of these species. With that said, we do not anticipate that they are indefinitely stable and would likely not persist until measurement at the UT Austin HRMS facility which we have used for this study. We are also unsure if this species, if present, would be stable to common HRMS ionization methods. We are interested in continuing to explore the detection and characterization of the BPin-O• radical and are designing future projects to gain more insight into this intriguing species.

4) Thanks to their response, I could understand that [1.1.1]propellane reacts with an alkyl radical so rapidly due to its large strain, while an alkyl radical reacts slowly with B₂pin₂. If this is the case, polymerization of BCP unit is prone to occur and therefore the use of 3 equivalents of B₂pin₂ seems essential. Any comments?

We thank reviewer for raising this insightful point! As the reviewer has surmised, oligomerization of BCP units, especially to the dimer (staffane) is a significant side product that must be controlled under many radical coupling conditions using [1.1.1] propellane. Key results from the Anderson group showed that staffane (BCP dimer) impurities can be significantly reduced by lowering the equivalents of [1.1.1]propellane included in a reaction, with one equivalent of [1.1.1]propellane providing an outstanding ratio of > 20:1 desired monomer coupling product/undesired staffane (please see supporting information of: *J. Am. Chem. Soc.* 2021, 143, 9729–9736. <https://doi.org/10.1021/jacs.1c04180> for more details). The importance of limiting [1.1.1] propellane equivalents for selective reaction was further demonstrated by the Molander group, who used 1.5 equivalents of [1.1.1]propellane and 3.0 equivalents of B₂pin₂ for their BCP boronate synthesis and reported no observation of oligomeric BCP units (*Nat. Chem.* 2022, 14, 1068–1077. <https://doi.org/10.1038/s41557-022-00979-0>).

From these key precedents, we arrived at our current reaction design to prevent formation of BCP oligomeric side products. Since [1.1.1] propellane is employed as the limiting reagent (1.0 equiv.) and B₂pin₂ is present in significant excess (3.0 equiv.), the reaction environment is dominated by the diboron species. Consequently, the high concentration of B₂pin₂ increases the probability of productive encounters between the bicyclo[1.1.1]pentyl radical and B₂pin₂. This concentration effect helps to kinetically favor borylation over undesired oligomerisation of the BCP unit and we do not observe any staffane or higher BCP oligomeric side products under our standard reaction conditions.

5) The chemoselective HAT of carboxylic acids by the Bpin-O radical is intriguing. It is likely that the formation of the complex between B₂pin₂, the acid, and DMSO is the key to this observed high selectivity. Alternatively, hydrogen bonding interactions between the acid and the oxygen atom of the boronate may be involved. I look forward to future research into this point.

We thank reviewer for raising this intriguing point. We also find the high chemoselectivity achieved by this unusual reagent combination is fascinating and the further exploration of this process and its applications in synthetic chemistry are ongoing projects in our research group.

We hope that our detailed revisions have satisfactorily addressed all of the points raised by the reviewer. We are extremely grateful for the reviewer's efforts to help us to improve the completeness and clarity of our study and their appreciation for the potential impacts of these findings.

Reviewer #3 (Remarks to the Author):

The authors sincerely addressed the reviewer's concern regarding the reaction mechanism by conducting additional experiments, which experimentally substantiated the proposed mechanism. Therefore, I consider that the manuscript can be accepted

We thank the reviewer for the thoughtful reassessment of our work. We are grateful for the reviewer's detailed and constructive comments that allowed us to obtain much stronger experimental support for the proposed mechanism and improve the clarity and preciseness of our manuscript.

Reviewer #2 (Remarks to the Author):

The authors sincerely responded to my additional comments and questions and all have been addressed properly. I enjoyed the discussion. I believe that this study would attract many chemists working in the related area, and therefore I strongly support the publication of the work in Nature Communications. I found another correction in Figure 1C. According to my memory and reference 34, the reaction shown below was reported in 2025.

We are grateful for the reviewer's enthusiasm for our work and excited to receive this recognition. We greatly appreciate the detailed suggestions that have helped improve the clarity and completeness of our study! We also thank the reviewer for pointing out the mistake and we have corrected **2020** to **2025** in Figure 1C.